# Levels of Total Mercury and Health Risk Assessment of Consuming Freshwater Stingrays (Chondrichthyes: Potamotrygoninae) of the Brazilian Amazon

**DOI:** 10.3390/ijerph20216990

**Published:** 2023-10-28

**Authors:** Adriano Teixeira de Oliveira, Paloma de Almeida Rodrigues, Alexandre Mendes Ramos Filho, Maria Fernanda da Silva Gomes, Ariany Rabello da Silva Liebl, Júlia Vianna de Pinho, Paulo Henrique Rocha Aride, Carlos Adam Conte-Junior

**Affiliations:** 1Animal Morphophysiology Laboratory, Academic Department of Teacher Training (DAEF), Federal Institute of Education, Science and Technology of Amazonas (IFAM), Manaus Centro Campus (CMC), Manaus 69020-120, AM, Brazil; gomesfernanda0807@gmail.com (M.F.d.S.G.); ny.rabello@gmail.com (A.R.d.S.L.); aride@ifam.edu.br (P.H.R.A.); 2Graduate Program in Animal Science and Fisheries Resources (PPGCARP), Faculty of Agricultural Sciences (FCA), Federal University of Amazonas (UFAM), University Campus, Manaus 69077-000, AM, Brazil; 3Center for Food Analysis (NAL), Technological Development Support Laboratory (LADETEC), Federal University of Rio de Janeiro (UFRJ), University City, Rio de Janeiro 21941-598, RJ, Brazil; paloma_almeida@id.uff.br (P.d.A.R.); alexandremrf11@gmail.com (A.M.R.F.); juliaviannaap@gmail.com (J.V.d.P.); conte@iq.ufrj.br (C.A.C.-J.); 4Laboratory of Advanced Analysis in Biochemistry and Molecular Biology (LAABBM), Department of Biochemistry, Federal University of Rio de Janeiro (UFRJ), University City, Rio de Janeiro 21941-909, RJ, Brazil; 5Graduate Program in Sanitary Surveillance (PPGVS), National Institute of Health Quality Control (INCQS), Oswaldo Cruz Foundation (FIOCRUZ), Rio de Janeiro 21040-900, RJ, Brazil; 6National Institute of Health Quality Control, Oswaldo Cruz Foundation, Rio de Janeiro 21040-900, RJ, Brazil; 7Graduate Program in Veterinary Hygiene (PPGHV), Faculty of Veterinary Medicine, Fluminense Federal University (UFF), Vital Brazil Filho, Niteroi 24220-000, RJ, Brazil; 8Graduate Program in Food Science (PPGCAL), Institute of Chemistry (IQ), Federal University of Rio de Janeiro (UFRJ), University City, Rio de Janeiro 21941-909, RJ, Brazil; 9Graduate Program in Chemistry (PGQu), Institute of Chemistry (IQ), Federal University of Rio de Janeiro (UFRJ), University City, Rio de Janeiro 21941-909, RJ, Brazil

**Keywords:** fish, riverine, elasmobranchs, metals, poisoning

## Abstract

Mercury is an element with potential risk to fish and those who consume it. Thus, this study aimed to determine the levels of total mercury (THg), carry out a health risk assessment related to the consumption of the freshwater stingrays *Potamotrygon motoro*, and determine the physical and chemical properties of the water where stingrays occur. Stingrays of the species *P. motoro* were obtained from the Amazon River, and samples of the animals’ musculature were collected to determine THg levels. Risk assessment was conducted using pre-established formulas of estimated monthly intake (EMI), maximum monthly intake rate (IRmm), and hazard quotient (HQ). Three population scenarios were evaluated, considering both sexes and differences between rural and urban areas. There was no relationship between weight and THg concentration nor between total length and THg concentration. Higher EMI values were observed in rural children; for the IRmm, male children had the lowest consumption levels. For the hazard quotient, there was a similarity between the three age groups when comparing the male and female sexes. In addition, the representatives of the rural area always had lower values than the urban area. Freshwater stingrays, like other elasmobranchs, can be crucial animal species because they act as sentinels in studies that assess harmful chemicals like mercury.

## 1. Introduction

Mercury (Hg) is an elemental metal frequently released into the environment due to human activities [1,2,3]. As a result of bioaccumulation and biomagnification along the trophic chain, Hg has been shown to have increasingly harmful effects on animals and humans [4,5]. As a result, Hg contamination impacts the income and food supply of riverine communities in the Amazon, which rely on extractive industries [6].

The presence of Hg and other toxic metals in water resources is associated with several factors, such as weathering, continental degassing, Hg leakage into water bodies, and industrial activities [7]. In the Amazon biome, the anthropic activities most related to contamination by toxic metals, including Hg, are deforestation, hydroelectric dams, and mining [8]. These activities have intensified in recent years, allied to the reduction in the performance of Brazilian environmental agencies [8].

Methylmercury (MeHg) is the most hazardous chemical form of mercury and has the most significant capacity to accumulate in the body through food consumption and adsorption on the skin [3]. Notably, the biota, the body’s predominant chemical species, absorbs roughly 95% of the ingested MeHg [3]. Thus, eating contaminated fish is of great concern, especially for riverside populations with high fish consumption. Furthermore, human intoxication via food consumption can trigger short- and long-term effects on the health of exposed people, causing neurological, reproductive, and immune damage [9].

In this sense, studies that evaluate Hg contamination levels are essential for health promotion and assessing local economic survival since fish is the primary source of animal protein of the population in the Amazon [10]. In the Amazon region, extractive activities are fundamental for the maintenance and survival of the local population, with fishing being the one that involves the most significant number of people [11]. This importance is due to the high consumption of fish, one of the largest in the world [11], many inland waters, and the increased diversity and abundance of fish in the region.

Traditionally, fish consumption in the Amazon is concentrated in teleost fish [11]. However, there has been an increasing interest in the meat of freshwater stingrays [12], which belong to the group elasmobranchs. Araújo [13] points out that, at the beginning of the 2000s, in the Amazon basin, places where freshwater stingrays were used as a food source were rare. However, in the second decade, Duncan, Inomata, and Fernandes [14] described the capture and consumption of freshwater stingrays in Baia de Marajó, Pará state. In this change in profile since 2005, there is an increase mainly in the capture of *Paratrygon aiereba* (Müller and Henle, 1841), *Potamotrygon motoro* (Müller and Henle, 1841), and *Plesiotrygon iwame* (Rosa, Castello, and Thorson, 1987) stingrays in the lower Amazon for commercial purposes [15], including food purposes.

The consolidation of freshwater stingray consumption in the last decade is due to the size of the animals, the interest in feeding people in the states of the northeast and southeast of Brazil, and as a way of reducing the natural stocks of fish species traditionally used as foods [12]. De Andrade et al. [16] recently described stingrays classified as *Potamotrygon* spp. sold in Amazon supermarkets; there is meat from the stingray *P. aiereba*, which is currently a species threatened with extinction in Brazilian territory. However, despite this growing interest in the consumption of freshwater stingray meat, information on levels of toxic metals, including Hg, still needs to be determined.

Thus, evaluating the toxic elements in freshwater stingrays is essential to determine the existing tissue content and how they may or may not harm human health. Therefore, the present study aims to assess mercury levels and a health risk assessment related to the consumption of freshwater stingrays *P. motoro*, as well as to determine the physical and chemical properties of the water where the stingrays occur.

## 2. Materials and Methods

The capture and procedures for the processing of biological material were previously authorized by the Chico Mendes Institute for Biodiversity Conservation (ICMBIO, State of Amazonas, Brazil, authorization no. 76.127-2), as well as authorization from the Ethics Commission in the Use of Animals (CEUA, Santa Catarina, Brazil, authorization no. 2019/010.02.0905) of the Federal Institute of Education, Science, and Technology of Amazonas (IFAM).

### 2.1. Study Area

The capture of the animals occurred in Andiroba Lake (Figure 1), located in the municipality of Manaquiri, Amazonas, which is part of the metropolitan region of Manaus and is bathed by the waters of the Amazon River. This lake is highly representative of the fishing activity in the area due to the diversity and abundance of fish and freshwater stingrays. Floodplain lakes in the Amazon region have mining activities that increase turbidity and promote mercury contamination and methylation, posing a direct threat to the lake’s ecological health [17]. The collections took place in August 2022 during the high-water period of the lower Amazon River.

### 2.2. Capture and Identification of the Animals

A total of 16 stingrays of the species *Potamotrygon motoro* (12 juveniles and 4 neonates) were captured with lures and fishing nets in activities that occurred during the night. The identification of the stingray species followed the identification key proposed by Lasso et al. [18]. The fish were classified according to the stage of ontogenic development, which was based on the disc width (DW, cm) and was determined with the use of a tape measure (neonates, DW ≤ 14.0 cm; young, 14.1 < DW ≤ 35.0 cm; subadults 35 < DW ≤ 40; and adults, DW > 40.0 cm); the classification followed the recommendations of Araújo [13]. Oliveira et al. [19] discussed the handling and confinement protocols used to measure the stingrays’ body weight using a portable scale.

After capture, the animals were anesthetized with eugenol (1 mg·L^−1^). They were killed via neural tube rupture [20], which consists of inserting a perforating cutting instrument at the base of the nervous system, allowing the gradual lethargy of the animal until sacrifice. Then, it was dissected by removing fillet portions for subsequent quantification of mercury (Hg) levels in relation to wet weight.

### 2.3. Mercury Quantification (Hg)

Blank values were always confirmed with values lower than 0.001 Hg (ng) before each sample analysis. The samples were then moved to the quartz trays and sterilized at 650 °C for 5 min in a muffle oven.

The THg present in *P. motoro* muscles was measured using atomic absorption spectrometry in a direct mercury analyzer (DMA-80, Milestone, Bergamo, Italy) after the previous calibration using a Hg solution (Sigma-Aldrich, São Paulo, Brazil). The calibration curve was composed of ten points (0.0, 0.5, 1.0, 2.0, 3.0, 5.0, 10.0, 20.0, 50.0, and 100.0 ng.g^−1^; 0–100 ng.g^−1^; y = 22.085 × 0.3217; r^2^ = 0.9999). The quality controls for the THg included blanks, replicates, and the reference material ERM-CE101-Trout muscle. Analytical parameter optimization was performed using the limit of quantification (LOQ) and limit of detection (LOD). The formulas were “LOD = (3 × SD)/slope” and “LOQ = (10 × SD)/slope”, where “SD” stands for the standard deviation of at least ten duplicates of the blank of the samples measured along this same curve divided by the slope of the corresponding calibration curve utilized in the analytical method. The calculated value was 0.0223 ± 0.0015, while the certified value was 0.02 ± 0.002, making the THg recovery value of accredited reference material 100%.

Three samples totaling 0.2517 ± 0.0010 g of muscle were weighed using a Shimadzu ATX224 analytical balance. The samples were placed in quartz trays and exposed to streams of oxygen heated to 160 °C for 1 min, 650 °C for 2 min, and 650 °C once more for 1 min at 3.1 atm. After drying, a gold amalgam trap was used to dissolve the Hg vapor, which was then heated and read. The detection was carried out at 253.7 nm, and the outcomes were provided in mg·kg^−1^. The manufacturer’s guidelines were used as the basis for the analysis (Milestone application note for mercury determination, Bergamo, Italy).

### 2.4. Risk Assessment Calculation

The pre-established formulas listed below were used to calculate the values of estimated monthly intake (*EMI*) (Equation (1)), maximum monthly ingestion rate (*IRmm*) (Equation (2)), and hazard quotient (*HQ*) (Equation (3)).
(1)$$EMI=\frac{C\times IR}{BW}$$
(2)$$IRmm=\frac{PTMI\times BW}{C}$$
(3)$$HQ=\frac{EF\times ED\times IR\times C}{RfD\times BW\times TA}$$

*EMI* stands for “estimated monthly intake”, “*IRmm*” for “maximum monthly ingestion rate”, “*IR*” for “ingestion rate”, “*C*” for “mercury concentration”, or “mg·kg^−1^”, and “*BW*” for “body weight”. The *IR* values were 3.41 kg per month in the urban area and 12.02 kg per month in the rural area. *HQ*: hazard quotient; *PTMI*: inadequate tolerated monthly intake (0.017 mg·kg^−1^ month^−1^). *RfD*: an estimate of a safe oral exposure level (Hg = 0.0001 mg·kg^−1^·day^−1^) [21]; *TA*: average exposure time for non-carcinogenic chemicals (*EF* × *ED*); *ED*: exposure duration (12 or 24 or 54 years); *EF*: exposure frequency (urban 48 days year^−1^ and rural 144 days·year^−1^).

The Joint Expert Committee on Food Additives (JECFA) set a monthly maximum intake limit, and the EMI calculation was used to confirm that this limit exceeded the average concentration of THg found in the ingested freshwater stingrays [22]. The weekly value of the provisional tolerable week intake (PTWI) (0.004 mg·kg^−1^·week^−1^) was converted to the provisional tolerated monthly intake (PTMI) (0.017 mg·kg^−1^·month^−1^). The IRmm specifies how many freshwater stingrays can be consumed simultaneously without exceeding the PTMI’s enforced cap. Subsequently, these data were used to calculate the hazard quotient (HQ) (Equation (3)). In the HQ equation, values above 1.0 represent potential harm to consumer health.

We conducted calculations based on the average weight of three age groups: children aged 12 years (boys 42 kg and girls 46 kg), young adults aged 24 years (men 72 kg and women 59 kg), and adults aged 54 years (men 78 kg and women 66 kg) to obtain a risk assessment for various age groups while also taking into account the differences between men and women. Furthermore, a distinction between urban and rural populations was made because, in the Amazon region, the riverine population consumes fish often and in great abundance [23].

### 2.5. Water Analysis

A multiparameter device (Orion Five Star, A329, United States) was used to analyze the water at the stingray capture locations and to measure the dissolved oxygen levels, pH, and temperature. A Secchi disk was also used to measure transparency. Using an enzymatic colorimetric kit (Alfakit, Acquacombo 1560, Brazil), the tales of alkalinity, hardness, nitrite, and nitrate were chosen.

### 2.6. Statistical Analyses

The Shapiro–Wilk normality test was used to evaluate the normal distribution of the data. To assess differences between mercury levels in neonates and juveniles, a Student’s t-test was used. Linear regression was employed to determine whether disc width and body weight influence mercury concentrations in freshwater stingrays. The program was R (R Core Team 2022^®^), and a significance level of 0.05 was adopted for assessments.

## 3. Results

Table 1 presents the biometric values of the captured *Potamotrygon motoro* specimens. No statistical differences were found between the concentration of THg in neonates and juveniles (Figure 2). There was no relationship between weight and THg concentration (Figure 3A, *p* = 0.721) nor between total length and THg concentration (Figure 3B, *p* = 0.872).

Higher EMI values were observed in children from rural areas compared to children from urban areas, with values about three times lower (Table 2). THg levels were lower than those recommended by the Brazilian regulatory agency (ANVISA) and international guidelines (Table 3). For the IRmm, male children presented the lowest consumption levels, with about half the value indicated for adult men (Table 3).

For the hazard quotient, there was a similarity between the three age groups when comparing the male and female sex (Table 4). In addition, the representatives of the rural area always had lower values than the urban area (Table 4).

The parameters of the water of the stingray capture sites, low dissolved oxygen values, and neutral pH were observed (Table 5). In addition, there were temperature aspects similar to those depicted in the waters of the Amazonian rivers and suitable visibility of the water in Andiroba Lake (Table 5).

## 4. Discussion

In addition to the physical and chemical properties of water, like pH, dissolved organic carbon, and the availability of potential methylation sites, environmental factors like animal size, life stage, trophic position in the food chain, and preferred substrate for habitat have also been linked to the dynamics of Hg in aquatic ecosystems [3,24].

In the Amazon, because there are at least three types of waters that have different physical and chemical properties, as well as the existence of environments that form in inland water bodies, such as floodplains and igapós, which cause differences in the diversity and abundance of fish for human consumption, the assessment of possible Hg poisoning should be conducted per basin, per season, and per fish species [25].

In a study conducted by Azevedo-Silva et al. [26] in which the level of mercury biomagnification and the trophic structure of the ichthyofauna of a remote lake in the Brazilian Amazon was evaluated, changes were observed according to trophic position, which was also correlated with biometric variables. However, in the present study, no statistical differences were observed between the two stages of development sampled (Figure 2) and neither in the biometric relationship with THg (Figure 3).

In freshwater stingrays, there is no difference in diet according to the stage of development [27], which is different from what is observed in other Amazonian fish, such as the tambaqui *Colossoma macropomum* (Cuvier, 1816), whose diet varies according to its stage of development [28]. However, freshwater stingrays of a larger size and weight may have higher values of THg than those observed in the present study. The disc width and body weight are factors of a more significant relationship with THg due to the growth of these freshwater elasmobranchs being of the positive allometric type [29].

Elasmobranchs are primarily predators, and many occupy top places in their respective food chains. Thus, contamination by toxic metals, including Hg, will be higher [29]. In this sense, the biomagnification of pollutants, such as Hg, in ingested prey is one of the reasons that predators tend to accumulate higher concentrations than other fish [30]. Elasmobranchs are also susceptible to the progressive accumulation of pollutants in various tissues of their bodies throughout their lives via processes of bioaccumulation and bioconcentration [30].

According to Shibuya [27], *Potamotrygon motoro* has a diet that ranges from insectivorous (insects) and carpophagous (decapods and fish) to generalist (mollusks, insects, and fish). When evaluating the stingray specimens following Shibuya [24], the animals in the present study had a diet that may vary from carcinomatous to generalist, including piscivorous and predatory fish. Bastos et al. [31] evaluated mercury levels in fish that are part of the stingray diet and observed values between 0.90 and 0.01 mg·kg^−1^ Hg for predatory fish and between 1.1 and 0.2 mg·kg^−1^ Hg for piscivores.

Elasmobranchs, including freshwater stingrays, are long-living animals; therefore, over decades and centuries, they have been susceptible to accumulating pollutants in the ecosystem [32]. On the other hand, some species of elasmobranch, such as *Prionace glauca* (Linnaeus 1758) and *Manta birostris* (Walbaum, 1792) [33], have a wider distribution and may not indicate the environmental characteristic of their capture site. Furthermore, freshwater stingrays have the attributes of being sedentary animals [19,20]; moreover, they have the biological aspect of burying themselves in the substrate [19,20], which is the central place of accumulation of Hg and, thus, may acquire more significant contamination by this element. In this way, these animals support the claim of being adequate bioindicator organisms, following the example of their marine relatives, sharks and rays [34].

The formation and accumulation of methylmercury have already been verified in tropical floodplain systems [35]. These aspects favor the bio-expansion and contamination of riverine populations in the Amazon, for which fish are the primary source of protein [35]. Furthermore, the place of capture of the stingrays used in the present study was a lake formed in floodplain areas. Due to the fluctuation in water levels, there may be an increase in Hg concentration through the environment itself.

The concentrations of THg detected in the investigated animals had lower values (Table 2) when compared to non-predatory animals (0.5 mg·kg^−1^ Hg) and top predators of the food chain (1.0 mg·kg^−1^ Hg) according to the Brazilian regulatory body [36] and international guidelines (0.5 mg·kg^−1^ Hg). Therefore, the contamination found in freshwater stingrays must come from contact with the substrate, and the contamination must be caused by mining activity in the Amazon, which has been intensifying in recent years [8].

The consumption of freshwater stingrays has yet to be common practice in the Amazon region [14]; however, Santos [12] indicates an increasing consumption trend of these elasmobranchs. In children of both sexes, it was observed that those from rural areas present EMI with values approximately three times higher than those found in children from urban areas. However, the EMI values were lower for young people and adults than those seen in children. In addition, values were about three times higher among residents in the rural area compared to those in the urban area.

In all scenarios evaluated, all rural and urban children, men, and women presented values above the maximum indicated PTMI (PTMI = 0.017 mg·kg^−1^ month^−1^). In addition, both men and women showed values above the maximum PTMI shown among the young and adults. Thus, only juveniles and adults in the urban area would be within the food security range to consume freshwater stingrays.

In this sense, due to biomagnification and bioaccumulation of Hg, consuming contaminated aquatic organisms is the main route of mercury exposure for traditional Amazon populations [37]. Thus, at all three levels evaluated, the riverine population demonstrates a dependence on fish from extractive activity.

Several studies in the Amazon region have shown that the fish consumed have higher Hg values than recommended, such as the works of Souza-Araújo et al. [38], Lino et al. [39], Vasconcelos et al. [40], and Vasconcelos et al. [41], in which these authors demonstrate that the basins of the Madeira River and the Tapajós River are of most significant concern due to mining activity in the area.

While evaluating health risks attributed to consuming fish contaminated with mercury in the Branco River basin in the Amazon, differences were observed between the urban and rural populations. The rural population presents higher doses of ingested Hg than the urban population [41], similar to the results obtained in the present study. The increased consumption of fish in the Amazon region (3.41 kg per month in the urban area and 12.02 kg per month in the rural area), especially for the riverine population and the values found for Hg concentrations, the consumption of freshwater stingray meat from the Amazon River requires attention and should not be consumed routinely, especially by children (males = 2.63 kg month^−1^; females = 2.86 kg month^−1^).

Fish is a type of food with high nutritional value that is due to its high protein content. In addition, many fish species are rich in minerals, vitamins, and polyunsaturated fatty acids that contribute to maintaining suitable health by preventing cardiovascular and neurological diseases [42]. However, despite the evidence of the benefits associated with fish consumption, the increasing contamination of the aquatic systems of the Amazon by mercury used in gold mining has led to discussions about the advantages and disadvantages of a diet rich in this type of animal protein.

An HQ > 1.0 indicates a risk to food health, so all groups evaluated in this study indicate a threat to human health, especially for children of either sex and young and adult women. For example, in a study that evaluated HQ in shark muscle, values of 1.7 were observed [43]. Therefore, it has also been recommended that humans avoid consuming shark meat (specifically muscle tissue), thus causing a health risk [43].

Alcala-Orozco et al. [44] performed biomonitoring of mercury in 14 species of fish of the Colombian Amazon and found an HQ above those allowed in 13 species. Their study ranged from 1.29 in *Leporinus agassizii* (Steindachner, 1876) to 24.13 in *Pseudoplatystoma tigrinum* (Valenciennes, 1840). Olivero-Verbel et al. [45] evaluated the human exposure and risks associated with mercury pollution in the Caqueta River in the Colombian Amazon and observed that all the fish species considered (N = 11) presented HQ values higher than recommended, with the species *Mylosoma duriventre* (Cuvier, 1818; HQ = 2.96) and *Cichla ocellaris* (Bloch and Schneider, 1801; HQ = 47.31) presenting lower and higher values, respectively.

For the parameters of the water of Andiroba Lake, which is bathed by the waters of the Amazon River, the dissolved oxygen values are low and following the studies conducted by Oliveira et al. [19], who evaluated waters of occurrence of three species of freshwater stingrays. The pH values demonstrate a water neutrality pattern characteristic of the whitewater of the Amazonian rivers [14]. The importance of the temperature and transparency of the water found in the present study indicates aspects similar to those portrayed for igapós and Amazonian lakes [19]. The colorimetric evaluations of alkalinity, hardness, nitrite, and nitrate found in the present study agree with the results described by Silva et al. [46], who evaluated the waters of the Amazon River.

## 5. Conclusions

The Hg levels found in the *P. motoro* stingray indicate values within the limits established for aquatic organisms. The EMI and IRmm values were considered adequate. However, the HQ showed values above ideal, indicating a risk for the Amazon population. These findings are caused by the high fish consumption rate in the region, especially by the people in rural areas. In this sense, evaluating Hg levels in aquatic organisms when considering HQ must be cautiously assessed since the values found may be within the established limits. Still, the HQ is above the permitted values. Therefore, Hg levels and high local fish consumption are equally important and must be considered.

Therefore, it is crucial to routine biomonitoring freshwater stingrays, as mining activity in the Amazon region has increased. It concluded that freshwater stingrays should be restricted, especially for people living in rural areas who rely mainly on fish for their animal protein. This will help prevent human health problems caused by mercury poisoning. Additionally, freshwater stingrays, like other elasmobranchs, are crucial animal species because they act as sentinels in studies evaluating harmful chemicals like mercury.

## Figures and Tables

**Figure 1 ijerph-20-06990-f001:**
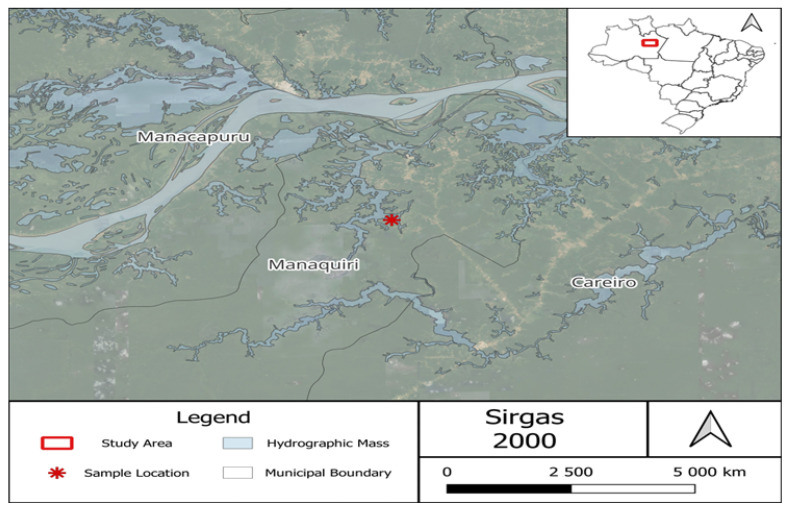
Andiroba Lake is in Manaquiri, lower Amazon River, Amazonas, Brazil. Collection point (3°33.398′ S, 60°28.503′ W).

**Figure 2 ijerph-20-06990-f002:**
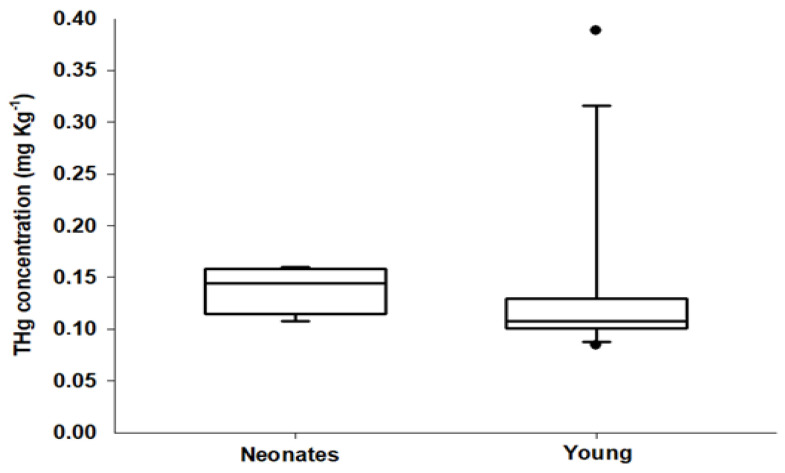
Box-plot representing comparative of THg concentration (mg·kg^−1^) in *Potamotrygon motoro* neonates and young in the Andiroba Lake, Amazon River, Amazonas, Brazil. The dot is a comparative analysis between stingray life stages.

**Figure 3 ijerph-20-06990-f003:**
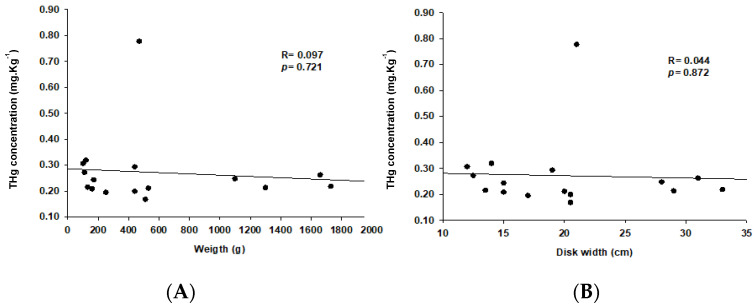
(**A**,**B**) Linear regressions between the biometric parameters and THg concentration in *Potamotrygon motoro* stingrays. (**A**) Regression between body weight and THg concentration. (**B**) Regression between disk width and THg concentration.

**Table 1 ijerph-20-06990-t001:** Mean and standard deviation of the biometrics of the freshwater stingray *Potamotrygon motoro* from the lower Amazon River, Amazonas, Brazil.

Developmental Stage	Disc Width (DW, cm)	Weight (g)
Neonates	12.50 ± 0.91	115.00 ± 12.91
Young	22.42 ± 6.23	673.85 ± 565.67

**Table 2 ijerph-20-06990-t002:** Estimated monthly intake (EMI) for *Potamotrygon motoro* freshwater stingrays from the Amazon River in Manaquiri, Amazonas, Brazil.

Species	EMI (mg·kg^−1^ Month)											
	Children				Young				Adult			
	Male		Female		Male		Female		Male		Female	
	U	R	U	R	U	R	U	R	U	R	U	R
*Potamotrygon motoro*	0.0221	0.0777	0.0201	0.0710	0.0129	0.0453	0.0157	0.0553	0.0119	0.0419	0.0140	0.0495

Body weight—Children (12 years): 42 kg male and 46 kg female; Young (24 years): 72 kg male and 59 kg female; Adults (54 years): 78 kg male and 66 kg female; U: Urban, R: Rural.

**Table 3 ijerph-20-06990-t003:** Maximum monthly ingestion rate (IRmm) and total mercury (THg) for *Potamotrygon motoro* freshwater stingrays from the Amazon River in Manaquiri, Amazonas, Brazil.

Species	IRmm (kg·month^−1^)												THg (mg·kg^−1^ w·w)
	Children				Young				Adult				
	Male		Female		Male		Female		Male		Female		
	U	R	U	R	U	R	U	R	U	R	U	R	
*Potamotrygon motoro*	2.63		2.88		4.51		3.69		4.88		4.13		0.272 ± 0.054

Body weight—Children (12 years): 42 kg male and 46 kg female; Young (24 years): 72 kg male and 59 kg female; Adults (54 years): 78 kg male and 66 kg female; U: Urban; R: Rural.

**Table 4 ijerph-20-06990-t004:** Results of the hazard quotient (HQ) evaluation in the three age groups of the consumers of freshwater stingrays sampled from the Amazon River, Manaquiri, Amazon, Brazil.

Species	HQ											
	Children				Young				Adults			
	Male		Female		Male		Female		Male		Female	
	U	R	U	R	U	R	U	R	U	R	U	R
*Potamotrygon motoro*	55.1	194.3	50.3	177.4	32.2	113.4	39.2	138.3	29.7	104.6	35.1	123.7

Children (12 years); Young (24 years); Adult (54 years); U, urban; R, rural.

**Table 5 ijerph-20-06990-t005:** Mean and standard deviation of the physical and chemical parameters of the water at the capture sites of *Potamotrygon motoro* stingrays in the lower Amazon River, Amazonas, Brazil.

Variable	Value
Dissolved oxygen (mg/L)	3.90 ± 0.72
pH	6.29 ± 0.15
Temperature (°C)	30.58 ± 1.86
Transparency (m)	1.80 ± 0.15
Alkalinity (mg/L)	47.50 ± 9.57
Hardness (mg/L)	35.0 ± 5.77
Nitrite (mg/L)	0.01 ± 0.01
Nitrate (mg/L)	0.10 ± 0.00

## Data Availability

All data generated during the analysis of this study are included in this article.

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
