# Peer review of "Levels of Total Mercury and Health Risk Assessment of Consuming Freshwater Stingrays (Chondrichthyes: Potamotrygoninae) of the Brazilian Amazon"

_ijerph, 2023, doi:10.3390/ijerph20216990_

Round 1
Reviewer 1 Report
Comments and Suggestions for Authors
This article is devoted to an urgent topic, similar studies are currently being conducted all over the world, and it is important to know in what quantities different types of aquatic inhabitants accumulate mercury in order to assess the risks to the health of the population that eats them and the risks to the ecosystem of which they are a part.
This article presents the results of a study of Stingrays of the species Potamotrygon motoro from the Amazon River. And although the scientific significance and reader interest in this topic is quite high, the material itself, in our opinion, requires improvement.
- In the introduction, it is necessary to clearly state the goals and objectives of the study.
- It is unclear from the description of the study area what is the source of the increased concentrations of mercury in the ecosystem of the river. The fishing significance of the water body is described, but there is no mention of contamination of bottom sediments, water from which mercury comes.
It is also necessary to note a limited number of samples, among which, as it becomes clear from the text, there are no adults.
- Mercury data are presented only in the figures, it is unclear whether mercury concentrations were determined in the dry or wet weight of hydrobionts.
- The drawings, especially Figure 1, have very small signatures and are hard to read.
- The risk assessment method used by the authors is quite applicable, its only limitation is the number of samples on the basis of which calculations are carried out.
- Conclusions, in our opinion, are poorly presented, they should be expanded - to summarize whether the goals have been achieved and whether the research tasks have been solved, what new scientific author's results have been obtained during the work, what should be continued, what tasks remain.
Author Response
1) In the introduction, it is necessary to clearly state the goals and objectives of the study.
Answer: we detail the objective, including determining the mercury level. We also specify which species of stingray was the objective of the research and the determination of the physical and chemical properties of the water in the places where the stingrays occur (see lines 90-92).
Due to the correction in the objectives of the work, we made adjustments to the abstract (see 31-35).
2) It is unclear from the description of the study area what is the source of the increased concentrations of mercury in the ecosystem of the river. The fishing significance of the water body is described, but there is no mention of contamination of bottom sediments, water from which mercury comes.
Answer: The information was included as requested (see lines 111-113). One cited work was included in the list of references.
3) It is also necessary to note a limited number of samples, among which, as it becomes clear from the text, there are no adults.
Answer: In this article, capturing adult individuals was impossible. It is worth mentioning that people traditionally fear the animals used in this study due to the presence of stingers and accidents. Furthermore, the locations where the animals are captured are challenging and captures always occur at night. We consider the sample number satisfactory, given the difficulties encountered.
4) Mercury data are presented only in the figures, it is unclear whether mercury concentrations were determined in the dry or wet weight of hydrobionts.
Answer: Samples were evaluated by wet weight (See line 137).
5) The drawings, especially Figure 1, have very small signatures and are hard to read.
Answer: Figure 1 has been replaced by another.
6) The risk assessment method used by the authors is quite applicable; its only limitation is the number of samples on the basis of which calculations are carried out.
Answer: The low sample number is explained by the difficulties in capturing and locating animals in a natural environment.
7) Conclusions, in our opinion, are poorly presented, they should be expanded - to summarize whether the goals have been achieved and whether the research tasks have been solved, what new scientific author's results have been obtained during the work, what should be continued, what tasks remain.
Answer: We thoroughly reviewed and added a new paragraph in the conclusion. Based on the comments proposed by the reviewer, we improved the work's title.
Additional comment: We would like to thank all the suggestions for revisions to the article. We consider them all relevant, and thank you for taking the time and knowledge to improve our article.
Recent work on trade in meat from an endangered species was included (see lines 87-90).
Reviewer 2 Report
Comments and Suggestions for Authors
The objective of this study is the quantification of total mercury levels in freshwater stingrays of the Brazilian Amazon to evaluate the human health risk associated with fish consumption.
The paper is clearly written, well referenced and the results are well documented and discussed.
I have only few specific remarks:
SECTION 2.4: I suggest to the authors to report in the manuscript the values of the ingestion rate (IR) used for the calculation of the estimated monthly intake (EMI).
LINE 155 and 159: The “maximum monthly ingestion rate” seems to be defined with two different acronyms: IRmm and MMIR.
LINE 163: Please define the variable “ED”.
LINE 168: Please define the acronym PTWI.
LINES 179-180: Authors report that in the study area urban population is less dense than the rural one. Can authors better explain this unexpected distribution of the population?
Author Response
The objective of this study is the quantification of total mercury levels in freshwater stingrays of the Brazilian Amazon to evaluate the human health risk associated with fish consumption.
The paper is clearly written, well referenced and the results are well documented and discussed.
Answer: Thanks.
I have only a few specific remarks:
1) SECTION 2.4: I suggest to the authors to report in the manuscript the values of the ingestion rate (IR) used for the calculation of the estimated monthly intake (EMI).
Answer: Information included. The IR values were 3.41 kg per month in the urban area and 12.02 kg per month in the rural area.
2) LINE 155 and 159: The “maximum monthly ingestion rate” seems to be defined with two different acronyms: IRmm and MMIR.
Answer: We corrected the acronyms and kept them all as IRmm.
3) LINE 163: Please define the variable “ED”.
Answer: Exposure duration (12, 24, or 54 years) (line 157). Information included.
4) LINE 168: Please define the acronym PTWI.
Answer: Called provisional tolerable week intake (line 181). An acronym description was inserted.
5) LINES 179-180: Authors report that in the study area urban population is less dense than the rural one. Can authors better explain this unexpected distribution of the population?
Answer: There was an error in writing this sentence. Therefore, we decided to exclude the phrase about the density of the urban and rural population.
Additional comment: We would like to thank all the suggestions for revisions to the article. We consider them all relevant, and thank you for taking the time and knowledge to improve our article.
Round 2
Reviewer 1 Report
Comments and Suggestions for Authors
The inscriptions in Figures 1, 2, 3 are still poorly readable.
In general, the comments have been eliminated as far as possible.